# Detection of Hyperdense Arterial Sign in Acute Ischemic Stroke with Dual-Energy Computed Tomography: Optimal Combination with X-ray Energy and Slice Thickness

Kyo Noguchi [1], Aki Kido [1], Norihito Naruto [1], Mariko Doai [1,*], Toshihide Itoh [2], Daina Kashiwazaki [3], Naoki Akioka [3] and Satoshi Kuroda [3]

[1] Department of Radiology, Graduate School of Medicine and Pharmaceutical Sciences, University of Toyama, Toyama 930-0194, Japan; kyo@med.u-toyama.ac.jp (K.N.); akikido@med.u-toyama.ac.jp (A.K.); naruto@med.u-toyama.ac.jp (N.N.)

[2] Department of CT Research & Collaboration, Siemens Healthineers, 1-11-1 Osaki Shinagawa-ku, Tokyo 141-8644, Japan; toshihide.itoh@siemens-healthineers.com

[3] Department of Neurosurgery, Graduate School of Medicine and Pharmaceutical Sciences, University of Toyama, Toyama 930-0194, Japan; dkashiwa@med.u-toyama.ac.jp (D.K.); akioka@med.u-toyama.ac.jp (N.A.); skuroda@med.u-toyama.ac.jp (S.K.)

* Correspondence: doaimari@med.u-toyama.ac.jp; Tel.: +81-76-434-7326

**Abstract:** Background: The hyperdense artery sign (HAS) in acute ischemic stroke (AIS) is considered an important marker of a thrombus on computed tomography (CT). An advantage of scanning with dual-energy CT (DECT) is its ability to reconstruct CT images with various energies using the virtual monochromatic imaging (VMI) technique. The aim of this study was to investigate the optimal combination of X-ray energy and slice thickness to detect HASs on DECT. Methods: A total of 32 patients with confirmed occlusion of the horizontal (M1) portion of the middle cerebral artery were included in this study. Modified contrast-to-noise ratio (modified CNR) analysis was used as a method for evaluating HASs in AIS. A region of interest (ROI) was set as an HAS, the M1 portion, and an approximately 2 cm diameter ROI was set as the background including the HAS and measured. CT images with X-ray energies from 40 to 190 keV, with increments of 10 keV, were reconstructed based on VMI with 1, 2, and 3 mm slice thicknesses. Results: The top five combinations of X-ray energy and slice thickness in descending order of the mean HAS-modified CNR were as follows: Rank 1, 60 keV-1 mm; Rank 2, 70 keV-1 mm; Rank 3, 60 keV-2 mm; Rank 4, 80 keV-2 mm; Rank 5, 60 keV-3 mm. Conclusions: Our study showed that the optimal combination to detect an HAS was 60 keV and a 1 mm slice thickness on DECT.

**Keywords:** hyperdense arterial sign; acute ischemic stroke; dual-energy CT; virtual monochromatic imaging

## 1. Introduction

Early stroke management provides the best outcomes for acute ischemic stroke (AIS) patients. In clinical practice, treatment of AIS patients includes intravenous thrombolysis (IVT) of the tissue plasminogen activator (rtPA) and mechanical thrombectomy (MT); both are indicated within a limited time window from symptom onset and require careful patient selection [1–3]. The recent American Heart Association/American Stroke Association (AHA/ASA) guidelines regarding the management of AIS emphasize the importance of MT treatment; it is recommended for patients in whom IVT is ineffective and should be initiated within 6 h of symptom onset [4]. Identifying the presence and location of thrombi on non-contrast computed tomography (NCCT) and CTA images is important when selecting patients with AIS for reperfusion therapy.

NCCT is widely employed to detect AIS because of its rapid performance, wide availability, and cost-effectiveness [5]. The current standard care protocol for AIS patients

includes the following [6]: first, NCCT to exclude intracranial hemorrhage and detect early ischemic changes; second, CT angiography (CTA) or CT perfusion (CTP) is practical and efficient to rapidly detect occluded vessels in AIS patients in order to perform MT treatment. CTA or CTP evaluation for AIS patients is an ideal approach although it poses risks of radiation and contrast exposure. Diffusion-weighted imaging (DWI) and magnetic resonance angiography (MRA) are not widely available and require time-consuming scanning that delays time-constrained treatment, and this is also inappropriate for unstable patients.

The hyperdense artery sign (HAS) in the middle cerebral artery (MCA) is considered the most important marker of a thrombus on NCCT [7,8]. Previous studies showed that an HAS in the MCA is present in 5 to 50% of cases [9,10]. The specificity of an HAS in the MCA to identify MCA occlusion is nearly 100%, whereas its sensitivity is low. However, such previous studies on HASs in the MCA were conducted using the conventional standard of more than 5 mm of thickness on NCCT [9,10]. Recently, it was reported in the literature that NCCT with a low slice thickness such as 1, 1.25, or 2.5 mm facilitated better identification of an HAS [11,12]. Dual-energy CT (DECT) has shown promising potential for various clinical applications [13–17]. It has the ability to generate several types of CT images from a single-acquisition dataset at high and low kV based on material decomposition (MD) algorithms. An advantage of scanning with DECT is the ability to reconstruct CT images with various energies using the virtual monochromatic imaging (VMI) technique. Regarding the detection of HASs by CT, the factors of X-ray energy and slice thickness may have an influence. The aim of this study was to investigate the optimal combination of X-ray energy and slice thickness on DECT to detect HASs in AIS patients.

## 2. Materials and Methods

### 2.1. Patients

Between May 2015 and August 2022, 93 consecutive patients with AIS were enrolled. Sixty-one patients were excluded for the following reasons: (1) detection of occlusion outside horizontal portion of middle cerebral artery (M1 portion), n = 43; (2) no detection of occlusion, n = 8; (3) DECT not performed on admission, n = 8; (4) insufficient range for analysis of DECT by motion artifacts, n = 2. A final total of 32 patients that underwent MT within 6 h after onset (16 males and 16 females; mean age ± standard deviation, 76 ± 12 years) were included in this study (Figure 1). Institutional review board approval was obtained with waived informed consent for retrospective analysis.

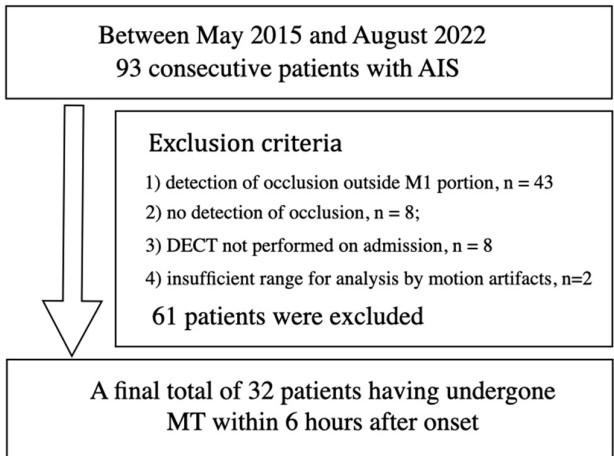

**Figure 1.** Flowchart for patient selection.

### 2.2. Dual-Energy CT

DECT examinations were performed using a dual-source DECT scanner (SOMATOM Force; Siemens Healthineers, Forchheim, Germany). Unenhanced head CT was performed on DE acquisition with two X-ray tubes operated at potentials of 80 and 150 kV with

tin filtration (Sn150 kV), which enlarges the effective energy separation between the two potentials. Scan parameters included a collimation width of 192 × 0.6 mm, rotation time of 1.0 s, and pitch of 0.7. Measurement of DE datasets (80 and Sn150 kV) was reconstructed with an increment of 1 mm and slice thickness of 1 to 5 mm. Virtual monoenergetic imaging (VMI) is based on two-material decomposition with two materials, such as iodine or bone and water. Once decomposition is complete, it allows computation of a simulated image of any X-ray using energies from 40 to 190 keV at 10 keV increments [17].

### 2.3. Modified CNR of HAS Measurement

Modified contrast-to-noise ratio (modified CNR) analysis was used as a method to evaluate HAS. A CT image with X-ray energies from 40 to 190 keV with increments of 10 keV was reconstructed based on VMI with 1, 2, and 3 mm slice thicknesses. A modified CNR of HAS in 48 CT images was measured in each patient, and a modified CNR of HAS in a total of 1536 CT images was measured in all 32 patients. A region of interest (ROI) was set as HAS of MCA in the horizontal portion (HAS portion), and an approximately 2 cm diameter ROI was set as the background including HAS (circle background) and measured (Figure 2).

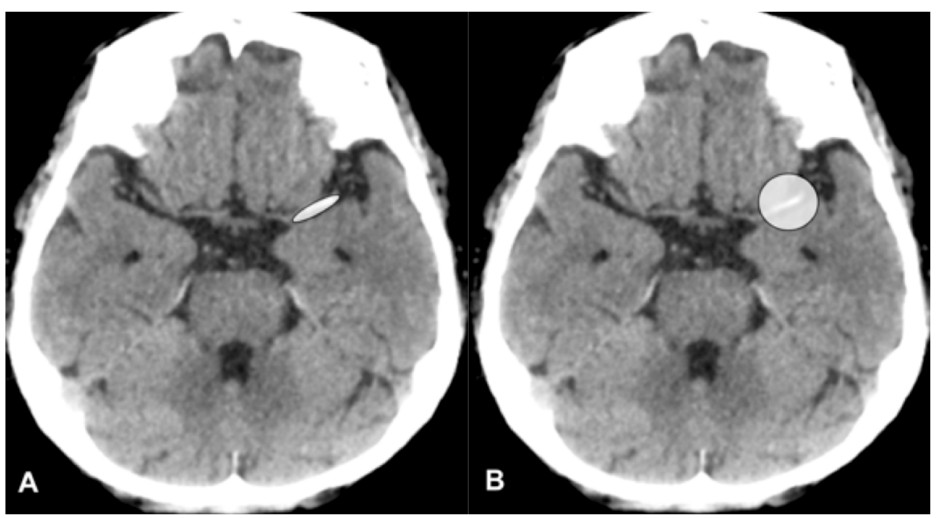

**Figure 2.** Region of interest (ROI). An ROI was set as HAS of MCA, horizontal portion (HAS portion) (**A**), and an approximately 2 cm diameter ROI was set as the background including HAS (circle background) (**B**).

The modified CNR of HAS was calculated as follows:

HAS-modified CNR = (HAS portion − Circle background)/standard deviation of Circle background.

### 2.4. Statistical Analysis

Statistical analysis of the mean modified CNR data of images from 40 to 190 keV at 10 keV increments was carried out by two-way analysis of variance (ANOVA) with Bonferroni correction. Statistical analysis of the mean modified CNR data of images between 1 and 2 mm, 1 and 3 mm, and 2 and 3 mm slice thicknesses was carried out using the *t*-test, respectively. All statistical calculations were carried out with PASW statistical software (ver. 29.0, SPSS, IBM, Chicago, IL, USA). A value of $p < 0.05$ was considered significant.

### 3. Results

Figure 3 shows all combinations of various X-ray energies and slice thicknesses in the modified CNR of an HAS. The modified CNR of an HAS upon combining 60 keV and a 1 mm slice thickness was highest (Figure 4). Table 1 shows the top ten combinations of X-ray energy and slice thickness in descending order of mean HAS-modified CNR: Rank 1, 60 keV-1 mm; Rank 2, 70 keV-1 mm; Rank 3, 60 keV-2 mm; Rank 4,

80 keV-2 mm; Rank 5, 60 keV-3 mm; Rank 6, 70 keV-2 mm; Rank 7, 90 keV-1 mm; Rank 8, 70 keV-3 mm; Rank 9, 80 keV-2 mm; Rank 10, 100 keV-1 mm. All three 60 keV CT combinations were included in the top five combinations (Table 1). The mean HAS-modified CNR at 40 keV was significantly lower than at 60 and 70 keV ($p = 0.0031$ and $0.012$, respectively). No other significant differences were observed. The mean HAS-modified CNR at a 1 mm slice thickness was significantly higher than at 2 and 3 mm ($p = 0.029$ and $0.003$, respectively) (Table 2) (Figure 5).

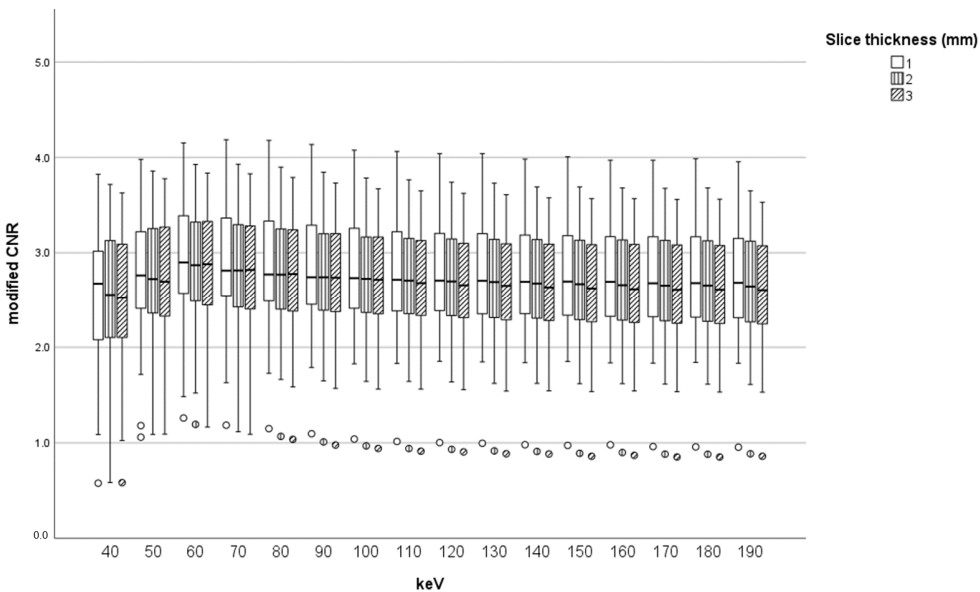

**Figure 3.** Boxplot of modified CNR (median, maximum, and minimum values) of HAS with the combination of energy and thickness.

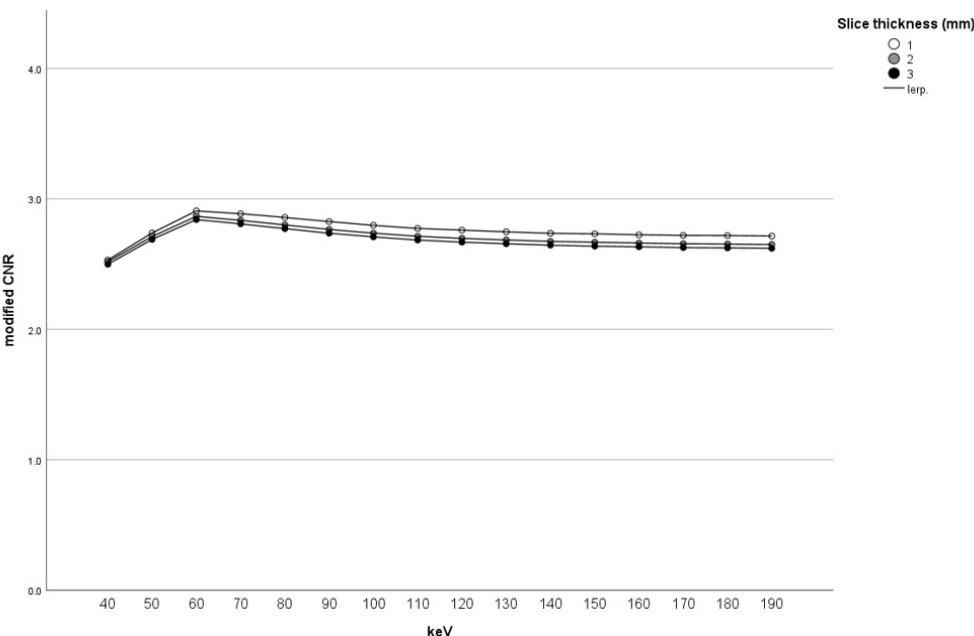

**Figure 4.** Mean modified CNR of HAS with the combination of energy and thickness.

**Table 1.** Top ten combinations with VMI settings and slice thicknesses in descending order of mean modified CNR of HAS on DECT.

| Rank | VMI Settings-Slice Thickness | Mean CNR of HAS | SD | 95% Confidence Interval Lower Limit to Upper Limit | |
|---|---|---|---|---|---|
| 1 | 60 keV-1 mm | 2.907 | 0.683 | 2.660 | 3.153 |
| 2 | 70 keV-1 mm | 2.886 | 0.668 | 2.644 | 3.126 |
| 3 | 60 keV-2 mm | 2.866 | 0.689 | 2.617 | 3.114 |
| 4 | 80 keV-1 mm | 2.857 | 0.653 | 2.594 | 3.056 |
| 5 | 60 keV-3 mm | 2.841 | 0.691 | 2.591 | 3.089 |
| 6 | 70 keV-2 mm | 2.835 | 0.671 | 2.592 | 3.077 |
| 7 | 90 keV-1 mm | 2.825 | 0.640 | 2.594 | 3.056 |
| 8 | 70 keV-3 mm | 2.808 | 0.672 | 2.565 | 3.050 |
| 9 | 80 keV-2 mm | 2.800 | 0.654 | 2.563 | 3.035 |
| 10 | 100 keV-1 mm | 2.797 | 0.632 | 2.568 | 3.024 |

**Table 2.** Statistical analysis of mean HAS-modified CNR of all tested VMI settings with each slice thickness.

| Thickness (I) | Thickness (J) | Mean CNR Difference (I-J) | Standard Error | *p*-Value | 95% Confidence Interval Lower Limit to Upper Limit | |
|---|---|---|---|---|---|---|
| 1.0 mm | 2.0 mm | 0.118 | 0.042 | 0.029 * | 0.007 | 0.229 |
| | 3.0 mm | 0.146 | 0.042 | 0.003 * | 0.035 | 0.257 |
| 2.0 mm | 1.0 mm | −0.118 | 0.042 | 0.029 * | −0.229 | −0.007 |
| | 3.0 mm | 0.027 | 0.042 | 10.000 | −0.083 | 0.138 |
| 3.0 mm | 1.0 mm | −0.146 | 0.042 | 0.003 * | −0.257 | −0.035 |
| | 2.0 mm | −0.027 | 0.042 | 10.000 | −0.138 | 0.083 |

* The mean HAS-modified CNR at 1 mm was significantly higher than at 2 and 3 mm.

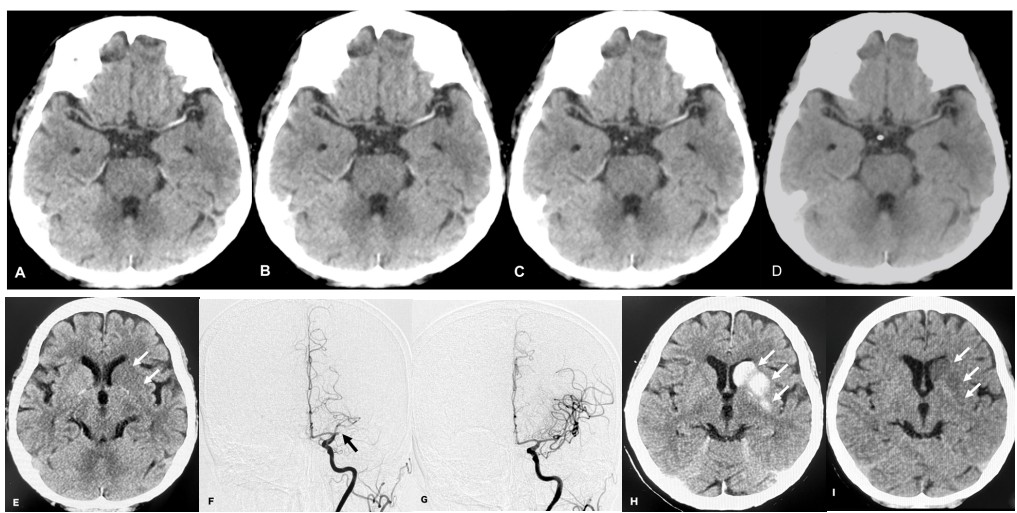

**Figure 5.** A patient with acute ischemic stroke 5 h after the ictus underwent MT. The 60 keV images (**A–D**) were reconstructed from VMI of DECT before MT. The 60 keV CT window width and window level were all 70 and 35 HU (hounsfield unit). HAS upon combination with 60 keV-1 mm slice thickness (**A**) offered the best contrast compared with 60 keV-2 (**B**), -3 mm (**C**), or -5 mm as a reference image (**D**). Simulated standard CT image reconstructed from DECT before MT showed early ischemic changes (arrows) in the left basal ganglia (**E**). Before MT, DSA identified left middle cerebral artery occlusion (arrow) (**F**). After MT, DSA showed that the middle cerebral artery occlusion was recanalized (**G**). Simulated standard CT reconstructed DECT after MT revealed hyper-density (arrow) in the left basal ganglia (**H**), and virtual non-contrast-image-reconstructed DECT after MT showed hypo-density (arrow) (**I**); therefore, this is considered to be iodine contrast material extravasation.

## 4. Discussion

In the present study, the mean and median modified CNR of an HAS with the combination of 60 keV and a 1 mm slice thickness were highest. The mean HAS-modified

CNR at 60 keV with 1, 2, and 3 mm CT images were included in the top five combinations (Table 1). Especially, the mean modified CNR of an HAS with the combination of 60 keV and a 1 mm slice thickness was highest (Figures 3–5). In terms of CT X-ray energy, 60 keV images were shown to be the most suitable to detect an HAS. This may be because about 60-70 keV CT images show the lowest noise [18]. Single-energy CT images (most at 120 kV) are equivalent to 70 keV CT images on DECT in terms of CT X-ray energy. Therefore, the optimal combination of 60 keV with a 1 mm slice thickness on DECT may be a possible substitute for single-energy CT (120 kV) with 1 mm to detect HASs.

In general, when detecting early ischemic changes before treatment, CT images with a slice thickness of more than 5 mm may be obtained. However, it is difficult to show the hyperdense sign using CT with such a large slice thickness due to the partial volume effect. The presence of HASs has been reported in only 17 to 50% of patients with the middle cerebral artery territory on thick-slice CT [9,10]. Two studies [11,12] examined the value of thin-slice CT reconstructed on multidetector CT. The results in both studies showed a marked rise in sensitivity to approximately 80–100%. In our study, the mean HAS modified CNR with a 1 mm slice thickness was highest, being significantly higher than that of 2 or 3 mm (Table 2).

Dual-energy CT (DECT) has shown promising potential for various clinical applications in AIS [13–17]. An advantage of scanning with DECT is the ability to reconstruct CT images with various X-ray energies using the VMI algorithm. In addition, it allows the differentiation of bone and calcification from other materials and also creates virtual non-contrast (VNC) by subtracting the iodine component from the image. One group has developed such a novel imaging technique called X-map to detect brain edema using DECT [19,20]. Another group [21,22] also developed a brain edema map based on DECT to detect brain edema in AIS.

NCCT has been playing a very important role in excluding intracranial hemorrhage and identifying AIS changes as well as HASs [5,6]. A single-acquisition non-enhanced DECT scan will provide much diagnostic information at the same time by (1) ruling out brain hemorrhage by virtual standard CT, (2) identifying the acute ischemic region by virtual standard CT and X-map, and (3) confirming HASs that indicate acute occlusion in large arterial vessels by 60 keV with a 1 mm slice thickness image.

Recent publications [2–4] regarding randomized controlled trials of MT showed promising results and may change the treatment protocol in the early stage of ischemic brain stroke treatment. The HAS is an important finding in the diagnosis of acute large arterial occlusion before MT treatment. If a definite HAS is detected, we can diagnose an occluded large vessel in AIS before this treatment. After MT treatment, hyperdense areas may represent either hemorrhage or contrast staining due to blood–brain barrier disruption, which can be reliably differentiated by VNC images on DECT (Figure 5I).

*Limitations*

Our study had several limitations. First, our results suggested that the mean modified CNR of an HAS with a 1 mm slice thickness was highest and significant. However, thin-slice CT images of less than 1 mm slice in thickness were not included in this study. The reconstructed images with slices thicknesses less than 1 mm may increase noise. There should be balance between the reduction in the partial volume effect and increasing image noise due to the thinner slice, which we aim to examine in the future. Second, modified CNR analysis was used as a method for evaluating HAS detection in AIS. A human observer study is the usual approach to evaluate task-based diagnostic performance under specific conditions. However, a human observer study was not conducted, because a total of 1536 CT images obtained from 32 patients had to be evaluated in this study. A human observer approach is time-consuming, and it is necessary to control the measurement carefully to ensure accurate results, which we aim to examine in the future. Last, this study had a retrospective design and included a relatively small sample size.

## 5. Conclusions

The mean modified CNR of an HAS with the combination of 60 keV and a 1 mm slice thickness was highest. Our study showed that the optimal combination to detect an HAS was 60 keV with a 1 mm slice thickness on DECT.

**Author Contributions:** Conceptualization, K.N.; methodology, K.N. and N.N.; software, N.N. and M.D.; validation, N.N., D.K. and N.A.; formal analysis, K.N., T.I. and M.D.; investigation, K.N., A.K., N.N., T.I. and M.D.; resources, K.N.; data curation, K.N., A.K. and M.D.; writing—original draft preparation, K.N.; writing—review and editing, K.N, A.K. and S.K.; visualization, D.K. and N.A.; supervision, K.N., A.K. and S.K.; project administration, K.N. and S.K.; funding acquisition, K.N. All authors have read and agreed to the published version of the manuscript.

**Funding:** This work was partly supported by JSPS KAKENHI Grant Number JP21K07669.

**Institutional Review Board Statement:** The study was conducted in accordance with the Declaration of Helsinki and approved by the Ethics Committee of University of Toyama (No. R2021029, date of approval 6 May 2021).

**Informed Consent Statement:** Patient consent was waived due to retrospective analysis.

**Data Availability Statement:** The data for this study are available from the authors.

**Conflicts of Interest:** The authors declare no conflicts of interest.

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
