# Peer review of "Detection of Hyperdense Arterial Sign in Acute Ischemic Stroke with Dual-Energy Computed Tomography: Optimal Combination with X-ray Energy and Slice Thickness"

_tomography, doi:10.3390/tomography10030028_

Round 1

Reviewer 1 Report

Comments and Suggestions for Authors

Dear Editor,

I recomend to reject the submitted paper. Below you can find my review:

Dear Authors,

Thank you for interesting manuscript. However, I believe that language editing and profound proof-reading would enhance the value of the manuscript and are necessary before resubmitting the paper.

I would like to find any comparison of the tested VMIs to linearly blended acquisition. It would allow to check if there are any differences in images resembling standard acquisition and tested VMIs. Furthermore, your proposed algorithm (discussion section) is based on other papers, which are not discussed properly. Please focus on your findings. Since you recommend a protocol utilizing DECT acquisitions have you considered radiation doses?

I have many minor comments:

1."First, NECT to exclude intracranial hemorrhage; Second, to perform NCCT to detect early ischemic changes" apart the typo in NECT abbreviation, this sentence suggest to perform NCCT twice.

2. “However, such previous studies on HAS in MCA were conducted using the conventional standard of more than 5-mm thickness on NCCT.” Reference(s) missing.

3. Reconstruction of images with different slice thickness is not a specific advantage of DECT. It is the advantage of multi-detector CTs available since many years.

4. Cosinder using abbreviation SD for standard deviation.

5. Sn150kV – can you explain?

6. “Virtual monoenergetic imaging (VMI) is based on 2MD with two materials, such as iodine or bone and water. Once decomposition has been completed, it allows computation of a simulated image at any X-ray energy from 40 to 190” – this is simply not true. VMI images are reconstructed based on differences in attenuation in different energies. What does 2MD means?

7. Usually, background ROIs are placed outside the target area (in ex. fat, air). Can you provide me reference where similar method was already used?

8. Figure 1 should be cited and placed in the same chapter – in MM or Results section. Now it is cited in MM and placed in Results. Placement of other figures and tables have to be corrected.

9. Explain Fig. 2 – what it exactly shows – mean, Confidence intervals, ranges?

 10. Fig. 4. Mention used  CT window settings. Remove unnecessary dot.

11. Table 1 – rather VMI settings than “variable X-ray energy” since these images are reconstructed virtually not acquired with those energies.

 12. Table 2 – is this a summary of the mean values acquired in all the tested VMI settings or specific energy (60 keV)?

 13. Discussion section is written chaotically, in first paragraph you should present your findings, then discuss it with current body of literature. You have summarized all your findings in one short sentence.

14. “VMI based on DECT allows computation of a simulated image at any X-ray energy from 40 to 190 keV every 10 keV.” – not true, you can set also smaller intervals. Probably yu made an assumption based on your software.

 15. “This may be because about 60-70-keV CT images show the lowest noise.” Usually high keV-VMIs show lowest noise. Can you provide any data supporting your claims?

Author Response

I believe that language editing and profound proof-reading would enhance the value of the manuscript and are necessary before resubmitting the paper. Your proposed algorithm (discussion section) is based on other papers, which are not discussed properly. Please focus on your findings. Since you recommend a protocol utilizing DECT acquisitions have you considered radiation doses?

We thank the reviewer for the valuable suggestions.

We have rewritten it as much as possible to improve the quality of English Language, including the part of Discussion that you mentioned.

The radiation dose of DECT was the same as that of single-energy CT in this study.

  1. "First, NECT to exclude intracranial hemorrhage; Second, to perform NCCT to detect early ischemic changes" apart the typo in NECT abbreviation, this sentence suggest to perform NCCT twice.

We thank the reviewer for the valuable comments.

We have rewritten the relevant part following your advice.

  1. “However, such previous studies on HAS in MCA were conducted using the conventional standard of more than 5-mm thickness on NCCT.” Reference(s) missing.

We are grateful for the valuable suggestions.

We have added references based on your advice.

  1. Reconstruction of images with different slice thickness is not a specific advantage of DECT. It is the advantage of multi-detector CTs available since many years.

We thank the reviewer for the important comments.

We have rewritten it following your advice.

  1. Consider using abbreviation SD for standard deviation.

We thank the reviewer for the valuable advice.

We have used the abbreviation SD for standard deviation as per the advice.

  1. Sn150kV – can you explain?

We appreciate the important question.

We have rewritten it following your advice.

  1. “Virtual monoenergetic imaging (VMI) is based on 2MD with two materials, such as iodine or bone and water. Once decomposition has been completed, it allows computation of a simulated image at any X-ray energy from 40 to 190” – this is simply not true. VMI images are reconstructed based on differences in attenuation in different energies.

We thank the reviewer for the valuable comments.

However, according to the study below by Yu, VMI used in our study is categorized as an image-based VMI and that is generated using the concept of a two-material decomposition algorithm, with two basis materials such as soft tissue and iodine for an enhanced scan or soft tissue and calcification for an unenhanced scan. The software we used in this study was commercially available from the vendor (syngo.via Dual-Energy software, Siemens Healthineers, Forchheim, Germany), which can generate VMI images at energy levels between 40 to 190 keV with 1-keV steps.

We have added the following paper as a reference:

Yu L, Christner JA, Leng S, Wang J, Fletcher JG, McCollough CH. Virtual monochromatic imaging in dual-source dual-energy CT: radiation dose and image quality. Med Phys 2011;38(12):6371-6379. doi: 10.1118/1.3658568

       What does 2MD means?

We thank the reviewer for the valuable question.

2MD is two-material decomposition.

We have rewritten it for clarification.

  1. Usually, background ROIs are placed outside the target area (in ex. fat, air). Can you provide me reference where similar method was already used?

We appreciate the valuable comments.

We agree that our method of placing ROI including the target area to measure the background image noise is probably not common. However, we believe this way accurately mimics our visual perception in a straightforward way. When we searched HAS, the most influential area to look at is not brain tissue or fat outside of our area of visual focus. However, we also agree with the reviewer’s concern. We have corrected "CNR" to "modified CNR" to help make readers aware that is not exactly the same as commonly understood CNR.

  1. Figure 1 should be cited and placed in the same chapter – in MM or Results section. Now it is cited in MM and placed in Results. Placement of other figures and tables have to be corrected.

We thank the reviewer for the valuable suggestion.

We have placed Figure 1 in MM following your advice.

  1. Explain Fig. 2 – what it exactly shows – mean, Confidence intervals, ranges?

We are grateful for the valuable question.

The boxplot shows the median, maximum, and minimum values.

  1. Fig. 4. Mention used CT window settings. Remove unnecessary dot.

We thank the reviewer for the important comment.

We have rewritten the relevant parts following your advice.

  1. Table 1 – rather VMI settings than “variable X-ray energy” since these images are reconstructed virtually not acquired with those energies.

We appreciate the valuable suggestion.

We have rewritten it based on your advice.

  1. Table 2 – is this a summary of the mean values acquired in all the tested VMI settings or specific energy (60 keV)?

We thank the reviewer for the valuable question.

This is a summary of the mean values acquired in all tested VMI settings.

We have rewritten it as per your advice.

  1. Discussion section is written chaotically, in first paragraph you should present your findings, then discuss it with current body of literature. You have summarized all your findings in one short sentence.

We are grateful for the valuable suggestion.

It has been revised following your advice.

  1. “VMI based on DECT allows computation of a simulated image at any X-ray energy from 40 to 190 keV every 10 keV.” – not true, you can set also smaller intervals. Probably yu made an assumption based on your software.

We thank the reviewer for the important comment.

However, virtual monoenergetic imaging (VMI) was generated at the 16 different energies between 40 to 190 keV with 10-keV steps using commercially available software (syngo.via Dual-Energy software, Siemens Healthineers, Forchheim, Germany).

  1. “This may be because about 60-70-keV CT images show the lowest noise.” Usually high keV-VMIs show lowest noise. Can you provide any data supporting your claims?

We are grateful for the important comment.

However, according to Figure 5 in the reference below, the image noise bottomed out at about 70 keV.

Yu L, Leng S, McCollough CH. Dual-energy CT-based monochromatic imaging. AJR Am J Roentgenol 2012;199 (5 Suppl):S9-S15. doi: 10.2214/AJR.12.9121

Reviewer 2 Report

Comments and Suggestions for Authors

Thank you for this interesting research! 

Attached you'll find my comments and suggestions. Overall, I think a minor revision is needed. 

Comments on the Quality of English Language

Author Response

Attached you'll find my comments and suggestions.

Overall, I think a minor revision is needed.

We thank the reviewer for the valuable comments.

We have revised it as much as possible based on your advice.

Reviewer 3 Report

Comments and Suggestions for Authors

-Short summary:

The authors aim to investigate the best compination of slice thickness and kev values at VMI to identify the velles " hyperdense artery sign" that is a marker of vascular trombosi in acute stroke

-Title:

"Detection of Hyperdense Arterial Sign" add "in acute ischemic stroke"

"n important marker of a thrombus" indicator of a thrombus presence

Add that this sign is assessed in ischemic stroke to better contestualize the study and help readers understand the topic at a glance

-Abstract:

wel-written. It summarized the main aspects of the study correctly.

-Introduction:

It provide a background for the study, but maybe some more information on the DECT can be valuable for the readers

"The recent AHA/ASA guidelines regarding the management of AIS  emphasize the importance of MT treatment (4). It is recommended for patients in whom IVT is ineffective. MT treatment is to be initiated within 6 hours of symptom onset." merge in a single phrase

"The hyperdense artery sign (HAS) in the middle cerebral artery (MCA) is considered 54 the most important marker of a thrombus on NCCT (7, 8). Previous studies showed that 55 HAS in MCA is present in 5 to 50% of cases (9, 10). The specificity of HAS in MCA to 56 identify MCA occlusion is nearly 100%, whereas its sensitivity is low. However, such pre- 57 vious studies on HAS in MCA were conducted using the conventional standard of more 58 than 5-mm thickness on NCCT. Recently, it was reported in the literature that NCCT with 59 a thinner slice thickness facilitated better identification of HAS (11, 12)" create a fluid section. the sentences are too short and disconnected

"Dual-energy CT (DECT) has shown promising potential for various clinical applica- 61 tions (13-17)." add relevant references (https://doi.org/10.3390/app13137653)

"The two elements of energy and slice thickness " I cannot get the concept of two elements

-Materials and Methods:

Well-presented

Add a flawchart showing patients selection, in section 2.1. Patients

-Results:

You can strenghten the value of the results providing a couple of clinical cases

-Discussion:

Clear and well-presented

-Strenght:

Topic of general interest

Article simple but clear and well-presented

Case series adequate to the study

-Liitations:

The topic is not new

A more precise timing of the CT execution (timing between symptoms onset and CT) coul represent a vluable information

Comments on the Quality of English Language

Merge some periods in fluid sections

Author Response

"Detection of Hyperdense Arterial Sign" add "in acute ischemic stroke"

We thank the reviewer for the valuable suggestion.

We have changed the title based on your advice.

  • "The recent AHA/ASA guidelines regarding the management of AIS emphasize the importance of MT treatment (4). It is recommended for patients in whom IVT is ineffective. MT treatment is to be initiated within 6 hours of symptom onset." merge in a single phrase

We are grateful for the valuable comments.

We have revised it as much as possible following your advice.

  • "The two elements of energy and slice thickness " I cannot get the concept of two elements

We have rewritten it as follows:

Regarding the detection of HAS by CT, factors of X-ray energy and slice thickness may have an influence.

  • Add a flawchart showing patients selection, in section 2.1. Patients

We thank the reviewer for the valuable suggestion.

We have added it following your advice.

  • You can strenghten the value of the results providing a couple of clinical cases

We thank the reviewer for the important suggestion.

Clinical information and other figures have been added to a case in Figure 5.

  • A more precise timing of the CT execution (timing between symptoms onset and CT) coul represent a vluable information

We are grateful for the comment.

A total of 32 patients who had undergone MT within 6 hours after onset were included in the study.

Round 2

Reviewer 1 Report

Comments and Suggestions for Authors

Dear Authors, 

Thank your your fast reply and revisions made. I believe that it has improved the quality of the manuscript. However, the used methodology remains my main concern regarding your manuscript. I'm afraid that the used methodology might have influenced the results and the results might be not comparable to the current body of literature. 

I have some minor comments:

1. AHA/ASA - please define abbreviation. 

2. "Recently, it was reported in the literature that NCCT with a thin slice thickness facilitated better identification of HAS" - define the thin slice thickness as described in the cited literature. 

Author Response

The used methodology remains my main concern regarding your manuscript. I'm afraid that the used methodology might have influenced the results and the results might be not comparable to the current body of literature. 

We appreciate the valuable comments.

Although our method of placing ROI may be not common, we believe this way accurately mimics our visual perception in a straightforward way. When we searched HAS, the most influential area to look at is not brain tissue or fat outside of our area of visual focus.

We have already corrected "CNR" to "modified CNR" to help make readers aware that is not exactly the same as commonly understood CNR. In this study with this ROI settings, modified CNR with thin slice thickness such as 1 mm showed the highest value. This result is in good agreement with previous results of other researchers (references 11, 12). Therefore, we believe that the modified CNR with this ROI setting is probably reasonable.

I have some minor comments:

1. AHA/ASA - please define abbreviation. 

We thank the reviewer for the valuable comment.

We have rewritten it following your advice.

2. "Recently, it was reported in the literature that NCCT with a thin 
slice thickness facilitated better identification of HAS" - define the 
thin slice thickness as described in the cited literature.

We thank the reviewer for the valuable comment.

We have rewritten it following your advice.

Thank you for your valuable suggestion.